

# Sweep Interpolation: A Fourth-Order Accurate Cost Effective Scheme in the Global Environmental Multiscale Model

Mohammad Mortezazadeh[1], Jean-François Cossette[2], Ashu Dastoor[1], Jean de Grandpré[1], Irena Ivanova[1], Abdessamad Qaddouri[2]

[1]Air Quality Research Division, Environment and Climate Change Canada, Dorval, H9P 1J3, Canada
[2]Meteorological Research Division, Environment and Climate Change Canada, Dorval, H9P 1J3, Canada

*Correspondence to*: Ashu Dastoor (ashu.dastoor@canada.ca)

**Abstract.** The interpolation process is the most computationally expensive step of the semi-Lagrangian (SL) approach for solving advection which is commonly used in numerical weather prediction (NWP) models. It has a significant impact on the accuracy of the solution and can potentially be the most expensive part of model integration. The sweep algorithm, which was first described by Mortezazadeh and Wang (2017), performs SL interpolation with the same computational cost as a third order polynomial scheme but with the accuracy of a fourth order interpolation scheme. This improvement is achieved by using two 3rd-order backward and forward polynomial interpolation schemes in two consecutive time steps. In this paper, we present a new application of the sweep algorithm within the context of global forecasts produced with Environment Climate Change Canada's Global Environmental Multiscale (GEM) model. Results show that the sweep interpolation scheme is computationally more efficient compared to a conventional fourth order polynomial scheme, especially evident for increasing number of several advected several passive tracers. An additional advantage of this new approach is that its implementation in a chemical and weather forecast models requires minimum modifications of the interpolation weighting coefficients. An analysis of the computational performance for a set of theoretical benchmarks as well as a global ozone forecast experiment show that up to 15% reduction in total wall clock time is achieved. Forecasting experiments using the global version of the GEM model and the new interpolation show that the sweep interpolation can perform very well in predicting ozone distribution, especially in the tropopause region where transport processes play a significant role.

## 1 Introduction

Following their development nearly 4 decades ago, SL schemes have been widely used by NWP models such as the Global Environmental Multiscale Model (GEM), the Integrated Forecasting System (IFS), and the Global Forecast System (GFS). (Robert, 1980; Staniforth and Côté, 1991; Ritchie et al., 1995; Girard et al., 2014; Husain and Girard, 2017; Husain et al., 2019; Mukhopadhyay et al., 2019). One of the main reasons for its popularity is the fact that it can use large time steps without suffering from the stability issues as the Eulerian methods, which are constrained by the Courant-Friedrich-Lewy condition (McDonald and Bates, 1987; Côté and Staniforth, 1988; Ritchie, 1988). SL schemes can be derived from an integral along the path that links a flow trajectory's departure point to its arrival point located on a fixed Eulerian grid (Smolarkiewicz and Pudykiewicz, 1992). From this perspective, the flow is considered as a group of discrete fluid particles



following their characteristic curves along the Lagrangian coordinate (Mortezazadeh and Wang, 2017). The SL scheme consists of two main steps. First, it finds the departure point by performing backward integration of the kinematic relationship describing the characteristic path of the fluid particles (McDonald and Haugen, 1992; Hortal, 2002). Second, an

interpolation scheme maps the data from the Eulerian grid to the position of the departure point (Williamson and Rasch, 1989; Purser and Leslie, 1991).

Choosing a suitable interpolation scheme plays an important role in the accuracy and computational cost of the SL approach (Yabe et al., 2001). Low order interpolation is susceptible to high dissipation error and as a result it may induce large conservation and shape preservation errors in the solution (Zerroukat, 2010; Mortezazadeh and Wang, 2017). Thus, a high

order interpolation scheme is preferable to reduce the dissipation error. Various high order interpolation schemes have been proposed to improve the accuracy of the SL approach such as cubic polynomial interpolation scheme, Hermite interpolation, Weighted Essentially Non-Oscillatory (WENO). (Shapiro and Hastings, 1973; Girard et al., 2014; Nakao et al., 2022; Petras et al., 2022). One of the common polynomial interpolation schemes that is used in weather and air quality forecasting systems is cubic interpolation scheme (Aires et al., 2004; Girard et al., 2014). This scheme can provide sufficient accuracy

for tracer transport and advection without being prohibitively computationally expensive.

Nonetheless, a significant amount of information from neighboring grid points is required to perform any high-order polynomial interpolation. For instance, a fourth-order-accurate 3D Lagrange interpolation scheme uses 64 grid points, i.e. 4 grids in each direction. Because of the significant computational cost of the high-order interpolation, the transport of tracers in atmospheric models can significantly affect the computational performance, especially when considering the evolution of

large number of tracers (Bradley et al., 2022). This issue is particularly important with the extensive development of operational Air Quality prediction systems which attempt to resolve a comprehensive set of physicochemical processes involving a growing number atmospheric chemical constituents (Im et al., 2015; Makar et al., 2015). Depending on the application, the number of tracers used in atmospheric air quality models can vary from few to hundreds, increasing computational cost of tracer transport. For example, Golaz et al. showed that in an atmospheric model with 40 tracers, 75%

of the dynamical core wall clock time is taken by the tracer transport solver (Golaz et al., 2019).

By contrast to the standard fourth-order-accurate interpolation, the sweep interpolation scheme can reduce the cost of the interpolation process by using fewer neighborhood grid points but keeping almost the same accuracy as fourth order interpolation. This scheme is based on the alternate use of third order backward and forward polynomial interpolation schemes (Mortezazadeh and Wang, 2017). This procedure is applied in two successive time steps, so that at the end of each

two time steps the truncation error is equal to the error of cubic interpolation scheme. Performance of the sweep interpolation scheme was investigated in 1D, 2D, and 3D benchmarks, a wave function and cavity flow in a 2D and 3D squares, respectively, and complete error analysis showing the accuracy of the scheme was reported (Mortezazadeh and Wang, 2017). In this paper, the sweep interpolation scheme is implemented into the Global Environmental Multiscale (GEM) model and its accuracy and computational performance in tracer transport are assessed. The paper is organized as follows. In the first

section we present the formulation of the sweep interpolation scheme. Next, three benchmarks are used to evaluate the





performance of sweep interpolation. Finally, the accuracy of sweep interpolation is evaluated by performing an ensemble of ozone forecasts over a 10 weeks period within the ECCC NWP global forecasting system based on the GEM model. Additionally, we investigate the performance of the new interpolation scheme by using different number of passive tracers. ECCC's air quality forecast model, GEM-MACH (Global Environmental Multiscale – Modelling Air quality and

CHemistry) (Zhou et al., 2021; Stevens et al., 2022), is an in-line chemistry-meteorology model based on the GEM model. The benefits of sweep interpolation scheme in reducing computational cost of the transport of chemical species in GEM-MACH will be tested in a future study.

## 2 Methodology

GEM is used for meteorological forecasting at all scales from 15 km to 2.5 km. It solves the elastic Euler equations using a

two-time-level Crank-Nicholson temporal discretization with SL treatment of the advection terms (Girard et al., 2014). The current version of GEM uses a finite difference discretization on a Yin-Yang grid (Qaddouri and Lee, 2011). Equations are discretized on an Arakawa C grid (Arakawa, 1988) in the horizontal and in the vertical direction, a hydrostatic-pressure coordinate is used and the equations are discretized on the Charney–Phillips grid. Tracer transport is accomplished by first solving the advection equation for a passive tracer and then by adding contributions from physics forcings in split mode. The

current interpolation scheme in GEM is fourth-order-accurate cubic Lagrange interpolation. It is used to calculate the variables at the departure point, as well as to perform the exchange of data on the boundaries of the two subgrids of the global Yin-Yang grid. In this study, we document the impact of using sweep interpolation for the advection of tracers as well as for the exchange of data between Yin and Yang sub grids in GEM. In the horizontal direction, spherical coordinates are used to solve the equations.

GEM solves a system of four prognostic equations, which include momentum (Eq. (1)), energy (Eq. (2)), mass conservation (Eq. (3)), and ideal gas law (Eq. (4)):

$$\frac{d\mathbf{V}}{dt} + f\mathbf{k} \times \mathbf{V} + \frac{1}{\rho}\nabla p + g\mathbf{k} = \mathbf{F}, \tag{1}$$

$$\frac{dT}{dt} - \frac{1}{\rho c_p}\frac{\partial p}{\partial t} = \frac{Q}{c_p}, \tag{2}$$

$$\frac{d\ln\rho}{dt} + \nabla.\mathbf{V} = 0, \tag{3}$$

$$\rho = \frac{p}{RT}, \tag{4}$$

where $\mathbf{V} = (\mathbf{V}_h, w)$, $T$, $p$, and $\rho$ are the velocity vector, temperature, pressure, and density, respectively. Here, $\mathbf{F}$ and Q are the source terms for friction and heat, respectively. $g$ is the gravitational force, and $c_p$ and $R$ are the thermodynamic parameters. The general form of nonlinear term in the prognostic equations and tracer transport equations can be written as

follows:



$$\frac{d\emptyset}{dt} = P, \tag{5}$$

where $\emptyset$ represents a prognostic variable or tracer scalars, and $P$ is the forcings. In GEM, by using the split approach, we solve Eq. (5) based on two steps. At first, we solve the pure advection equation and calculate the intermediate solution $\emptyset_A^*$.

$$\frac{d\emptyset}{dt} = \frac{\partial\emptyset}{\partial t} + u\frac{\partial\emptyset}{\partial X} = 0, \tag{6}$$

where $X = (x, y, z)$ and $t$ specify spatial and temporal coordinates. Then by using semi-Lagrangian scheme, Eq. (6) is written as:

$$\emptyset_A^* = \emptyset_D, \tag{7}$$

Here, $A: (X, t)$ and $D: (X - \Delta X, t - \Delta t)$ correspond to the arrival and departure positions of flow trajectories, respectively (see Figure 1).

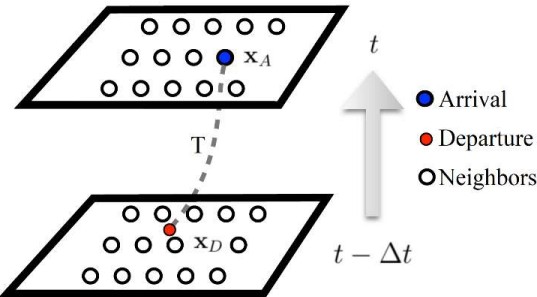

Figure 1. Procedure of time marching in semi-Lagrangian scheme.

In the second step, we add the forcing term to the intermediate solution to find the results at the present time step:

$$\emptyset_A = \emptyset_A^* + \Delta t P, \tag{8}$$

In the semi-Lagrangian method, the most challenging part is calculating the value of the variables at the departure point $X_D = (x_D, y_D, z_D)$. To accomplish this, first we need to find the position $X_D$ by moving back along the flow trajectories $T$ at $X_A$:

$$X_D = X_A - \int_{t-\Delta t}^{t} V(X(\tau), \tau)\, d\tau, \tag{9}$$

By using an averaging procedure for the integral of velocity on the right-hand-side of the Eq. (9), this equation can be written as follows:

$$X_D = X_A - BV_A + (1 - B)V_D, \tag{10}$$

where $B \geq 0.5$ is an off-centering weight. Because the position of the departure point ($X_D$) is not necessarily on the grid points, use of an interpolation scheme is unavoidable to transfer the data from Eulerian grids to the position of the departure point (Williamson and Rasch, 1989; Purser and Leslie, 1991). Lagrange interpolation is used in GEM.

$$\emptyset_A^* = \sum_l w_l \emptyset_l^{t-\Delta t}, \tag{11}$$



where $l$ represents the index of neighbor grid points surrounding the departure point and $w_l$ are the interpolation weights. In the current version of GEM, cubic interpolation is available. In this method, for doing the interpolation, 64-neighbor cells,

i.e. 4 grids in each direction, are considered to achieve the fourth order of accuracy in space. In the following, we explain the Mortezazadeh and Wang (2017) method to generate fourth order of spatial accuracy by using only 27 neighbor cells, which can lead to a faster interpolation process.

Here, for the sake of simplicity, we discretize the equations and investigate the numerical approach based on 1D advection $(X \rightarrow x)$ and uniform grid with the grid size of $\Delta x$ (see Figure 2).


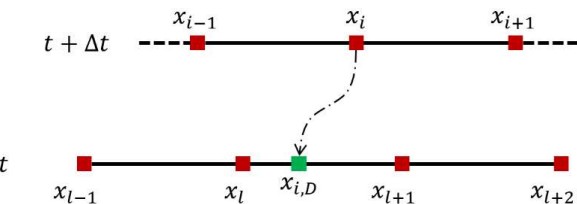

Figure 2. 1D computational stencil diagram of neighborhoods near the departure point.

As mentioned before, cubic Lagrange interpolation scheme uses 4 grid neighborhoods ,on 1D, to calculate $\emptyset_D$ at the departure point $x_{i,D}$, i.e. $x_{l-1}$, $x_l$, $x_{l+1}$, and $x_{l+2}$. Here $i$ is the index of arrival cells or Eulerian grids. Note that here using the

Lagrange interpolation scheme, we can construct a prediction function $\widehat{\emptyset}(x_{i,D})$ from the exact solution of $\emptyset(x_{i,D})$ within the cell $[x_l, x_{l+1}]$. Error in cubic interpolation is as follows:

$$E_{Cubic} = \emptyset(x_{i,D}) - \widehat{\emptyset}(x_{i,D}) = \frac{1}{4!}\emptyset^{(4)}(x_{i,D}) \times \prod_{l-1}^{l+2}(x_{i,D} - x_l) = O(\Delta x^4), \tag{12}$$

Sweep interpolation is an integration of two 3rd-order backward and forward interpolation steps that are used successively in two time steps. In the following, we show how sweep interpolation can achieve almost the same accuracy as cubic interpolation by using fewer neighborhoods. Equation (13) shows the numerical errors in 3rd-order backward $(E_B)$ and

forward $(E_F)$ polynomial interpolation scheme:

$$E_B = \frac{1}{3!}\emptyset^{(3)}(x_{i,D}) \times \prod_{l}^{l+2}(x_{i,D} - x_l) = -O(\Delta x^3),$$

$$E_F = \frac{1}{3!}\emptyset^{(3)}(x_{i,D}) \times \prod_{l-1}^{l+1}(x_{i,D} - x_l) = O(\Delta x^3), \tag{13}$$

From Eq. (13) we will find that the numerical error for the two interpolation steps is almost the same but with opposite sign. While use of either backward or forward interpolation scheme after two time steps can generate $E_{B,2\times\Delta t} = 2 \times E_B$, using 3rd-order backward and forward interpolation successively in two time steps can eliminate the 3rd-order numerical errors $O(\Delta x^3)$ and convert the total interpolation error to $O(\Delta x^4)$.



$$E_{2 \times \Delta t} = E_B + E_F = O(\Delta x^4), \tag{14}$$

Although the analysis of the numerical errors is shown by assuming a constant $\Delta x$ (uniform grids) in the above equations, it is also valid for non-uniform grids (Mortezazadeh and Wang, 2017). In the next section, we will see how this interpolation approach can control the error after every two-time steps in GEM.

In this study, sweep interpolation is used not only for semi-Lagrangian advection, but also for exchanging the data between Yin and Yang grids in GEM. In the following, we investigate the performance of the model by solving three benchmarks and

an ozone prediction case.

### 3 Validation

### 3.1 2D vortex

The 2D vortex is a conventional benchmark which is used here to compare the accuracy of the sweep interpolation with the cubic interpolation. This case consists of a scalar field that is deformed by two static vortices centered at the geographical

poles (Nair and Machenhauer, 2002). Here, maximum value of Courant number is 0.85426 ($(\Delta t = 7200 \, [s])$). By considering $(\lambda, \varphi)$ as the rotated coordinate system, North Pole location: $\lambda_0, \varphi_0$, and an angular velocity: $\omega$, we can write the rotation of the scalar field as:

$$\frac{d\lambda}{dt} = \omega, \tag{15}$$

$$\frac{d\varphi}{dt} = 0,$$

The normalized tangential velocity of the vortex is defined as:

$$V_t = \frac{3\sqrt{3}}{2} \text{sech}^2(r) \tanh(r), \tag{16}$$

where $r = r_0 \cos \varphi$ is the radius of the vortex and $r_0$ is a constant. Then, the angular velocity $\omega$ is defined as:

$$\omega(\varphi) = \begin{cases} 0 & if \ r = 0 \\ \frac{V_t}{r} & if \ r \neq 0 \end{cases}, \tag{17}$$

The analytical solution of this case at time $t$ is available in the reference (Nair and Machenhauer, 2002). Figure 3(a) shows error in the tracer at day 12. It can be seen that the error distribution inside the domain is almost similar for both schemes.



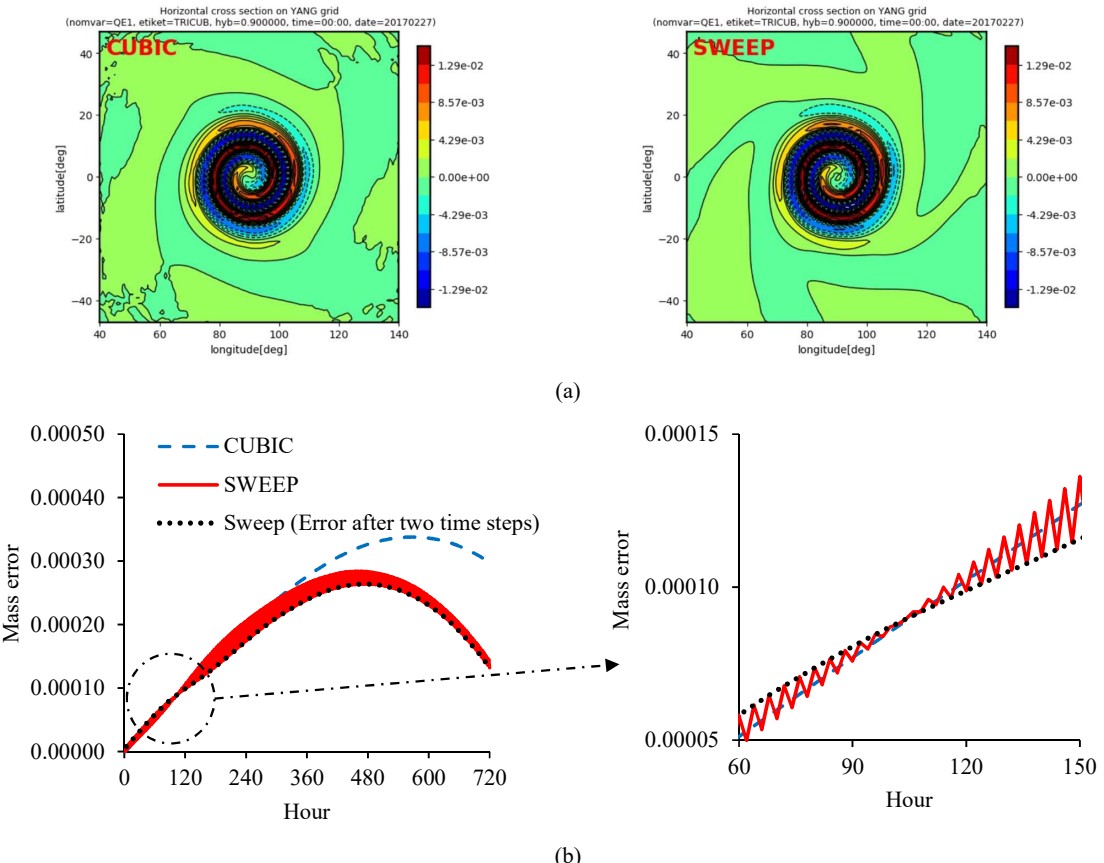

Figure 3. 2D vortex: (a) Errors in tracer distribution at day 12, (b) Mass conservation error (in %) over 1 month period.

Figure 3(b) shows the mass error over 1 month. This figure shows an oscillation in mass error, calculated by sweep interpolation, which comes from the inherent behavior of sweep interpolation to control the growth of numerical error over the iterations. The black dotted line in the figure 3(b) shows the mass error after each two time steps. This line represents the impact of sweep interpolation to reduce the error. The same behavior has been seen in the other test cases (see next sections).

### 3.2 Hadley-like meridional circulation

The Hadley-like Meridional Circulation test case is used to emphasize the solution of horizontal-vertical coupling (Ullrich et al., 2013). The tracer field consists of a single layer, which is deformed over the duration of 24 hours. At the end, the tracer field returns to its original configuration. Meridional and vertical velocity fields for this test case are specified as:

$$u(\lambda, \varphi, z, t) = u_0 \cos \varphi, \tag{18}$$



$$v(\lambda, \varphi, z, t) = \frac{aw_0\pi\rho_0}{Kz_{top}\rho}\cos\varphi\,\sin(K\varphi)\cos(\frac{\pi z}{z_{top}})\cos(\frac{\pi t}{\tau}), \tag{19}$$

$$w(\lambda, \varphi, z, t) = \frac{w_0\rho_0}{K\rho}(-2\sin(K\varphi)\sin\varphi + K\cos\varphi\cos(K\varphi))\sin\left(\frac{\pi z}{z_{top}}\right)\cos\left(\frac{\pi t}{\tau}\right), \tag{20}$$

where $\rho_0$, $K = 5$, $z_{top} = 12000\,[m]$, $w_0 = 0.15\,[m\,s^{-1}]$, $u_0 = 40\,[m\,s^{-1}]$, $a$ and $\tau = 86400\,[s]$ are the density at the surface, number of overturning cells, height position of the model top, reference vertical velocity, reference zonal velocity, reference earth radius, and period of motion. The maximal Courant number $CFL = 5.0$ ($\Delta t = 3600\,[s]$).

Figure 4(a) presents the meridional cross sections of the errors in the tracer solution after a 24 hours integration period. The results show that sweep interpolation generates a solution (right panel) that is similar to the one generated with the fourth-order-accurate cubic interpolation (left panel). Figure 4(b) shows the evolution of the error in the mass conservation (in %) over the same 24 hour period corresponding to both interpolations. The evolution in both solutions remain almost the same for both sweep and cubic interpolations. Note that here we can see the oscillations in error for sweep interpolation scheme.

As explained in the previous section, this shows how sweep interpolation can control the error growth over each two-time steps.

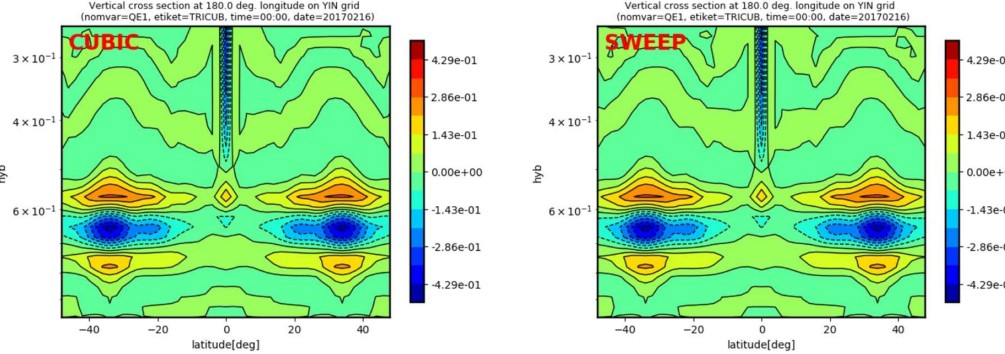





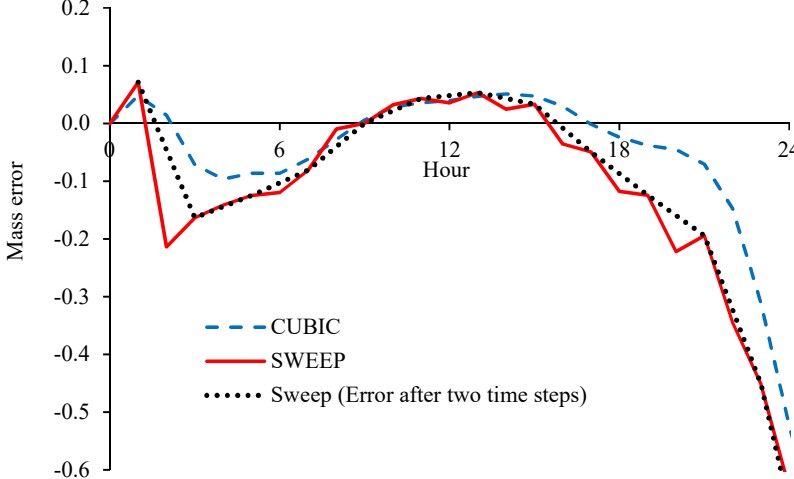

Figure 4. Hadley-like Meridional Circulation: (a) Errors in tracer at day 1, (b) Mass conservation error (in %) over 1 day.

**3.3 Atmospheric methane-like tracer**

In this test case, we compare 48 hour forecasts of atmospheric methane ($CH_4$) like passive tracer (without chemical
productions and sinks) using different interpolation schemes, sweep interpolation and cubic interpolation. These experiments
were performed with a 30 minute time step at 84 levels resolution in the vertical direction to give a maximal courant number
of 4.7. The grid distribution along the vertical direction is non-uniform. The horizontal grid resolution is 115 km. Figure 5(a)
presents meridional cross sections of $CH_4$ at the end of day 1. Solutions from both interpolators look qualitatively the same,
and the sweep interpolation provides acceptable results in comparison with cubic interpolation. Figure 5(b) shows the mass
error over 24 hours. It shows that both cubic and sweep interpolations could control the error and keep its range below
0.005% after 48 time steps of simulation.





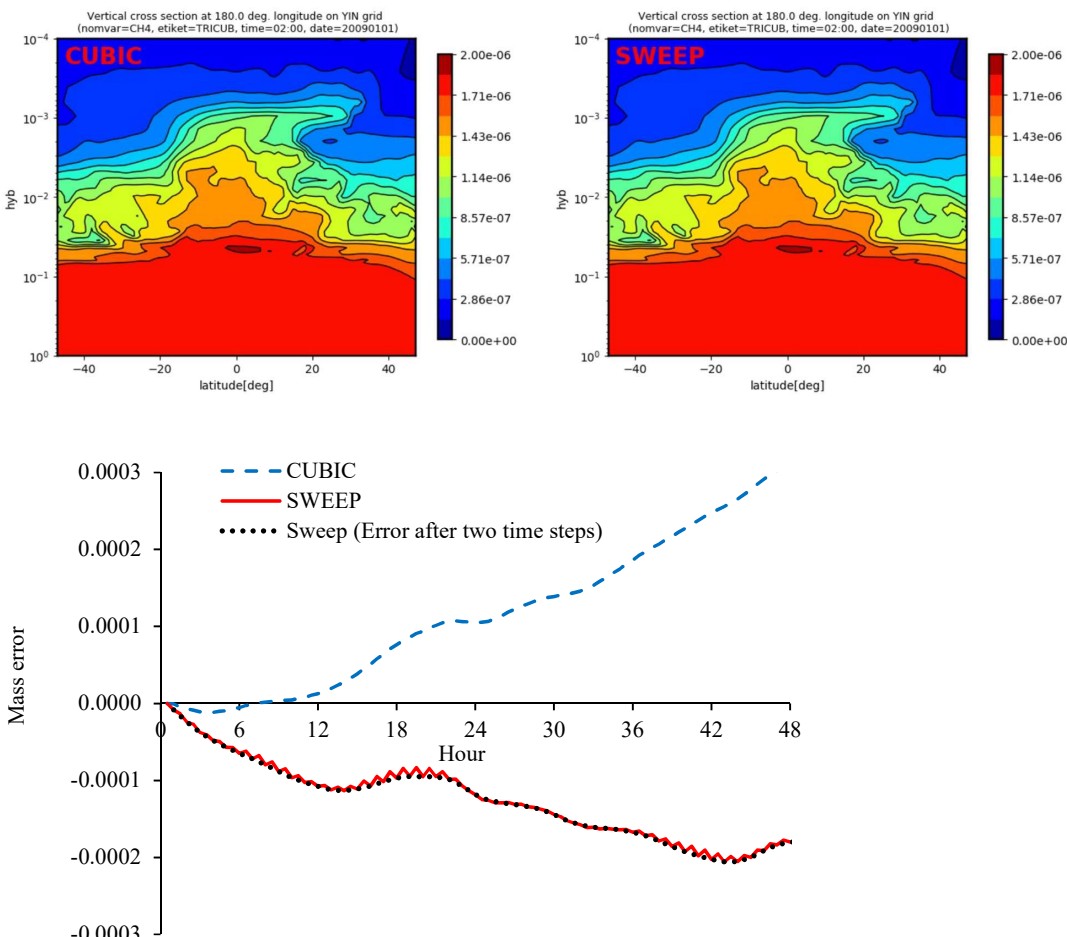

Figure 5. CH4 Forecasts: (a) Meridional cross sections at 180 degree longitude at the end of day 1, (b) Normalized mass error (in %) over 2 day forecast.

Although sweep interpolation could better control the mass error growth over the simulation time for this case, it is not
necessarily expected to perform better in all cases. Based on our discussion in the previous section, we expect sweep interpolation to provide almost the same accuracy as cubic interpolation.



### 3.4 Medium range forecasts of ozone

In the UTLS region, ozone is mainly driven by transport processes which makes it a valuable tracer for assessing the impact of the new interpolation scheme on the transport of chemical constituents. The implementation of sweep in the ECCC GEM NWP model allowed evaluation of its impact on the ozone predictability in the context of an operational NWP system. In this experiment, an ensemble of 10-day ozone forecasts were launched every 12 hours from June 13th to August 31st 2019 using the global version of the GEM NWP model. These forecasts were performed at 15 km horizontal resolution on 84 vertical levels with a lid at .1 hPa and using a 450 seconds timestep. The model included a prognostic representation of stratospheric ozone based on a linearized chemistry scheme which did not interact with the radiation (de Grandpré et al., 2016). Ozone forecasts have been evaluated at different lead-times against operational analyses generated by the Global Deterministic Prediction System (Charron et al., 2012). This evaluation was done throughout the tropospheric and stratospheric regions in various areas including Northern/Southern extratropics, North America, Asia, Arctic and Antarctic. The results from two interpolations, i.e. cubic and sweep interpolations, at different locations over the whole earth experiments demonstrate a good agreement between sweep interpolation and cubic interpolation. The two main metrics of the statistical accuracy, bias and standard deviation, are used to show the differences between the two experiments. In the following figures (Figure 6 and 7), solid and dashed lines represent the standard deviation and bias metrics, respectively. Figure 6 represents the average of all 10 day lead-time forecasts in the simulation time period. This figure shows that the standard deviation for both experiments are almost the same, but some minor deviations can be seen in the bias. The bias for ozone variation along the vertical direction is similar for both experiments in the lower part of troposphere (>500 hPa) and in the upper stratosphere (<50 hPa). The smaller concentration of ozone in these two regions accounts for the absence of sharp gradients in the vertical error profiles. The large maximum in the standard deviation around 250 hPa is associated with the transport of ozone across the tropopause. This process is sensitive to the choice of interpolation scheme and the results highlight its impact on stratosphere-troposphere exchange processes throughout the entire upper troposphere and lower stratosphere region between 400 and 50 hPa. In this region, a minor deviation of bias error can be seen between sweep interpolation and cubic interpolation, but this deviation is small and not significant (see Figure 6).

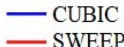





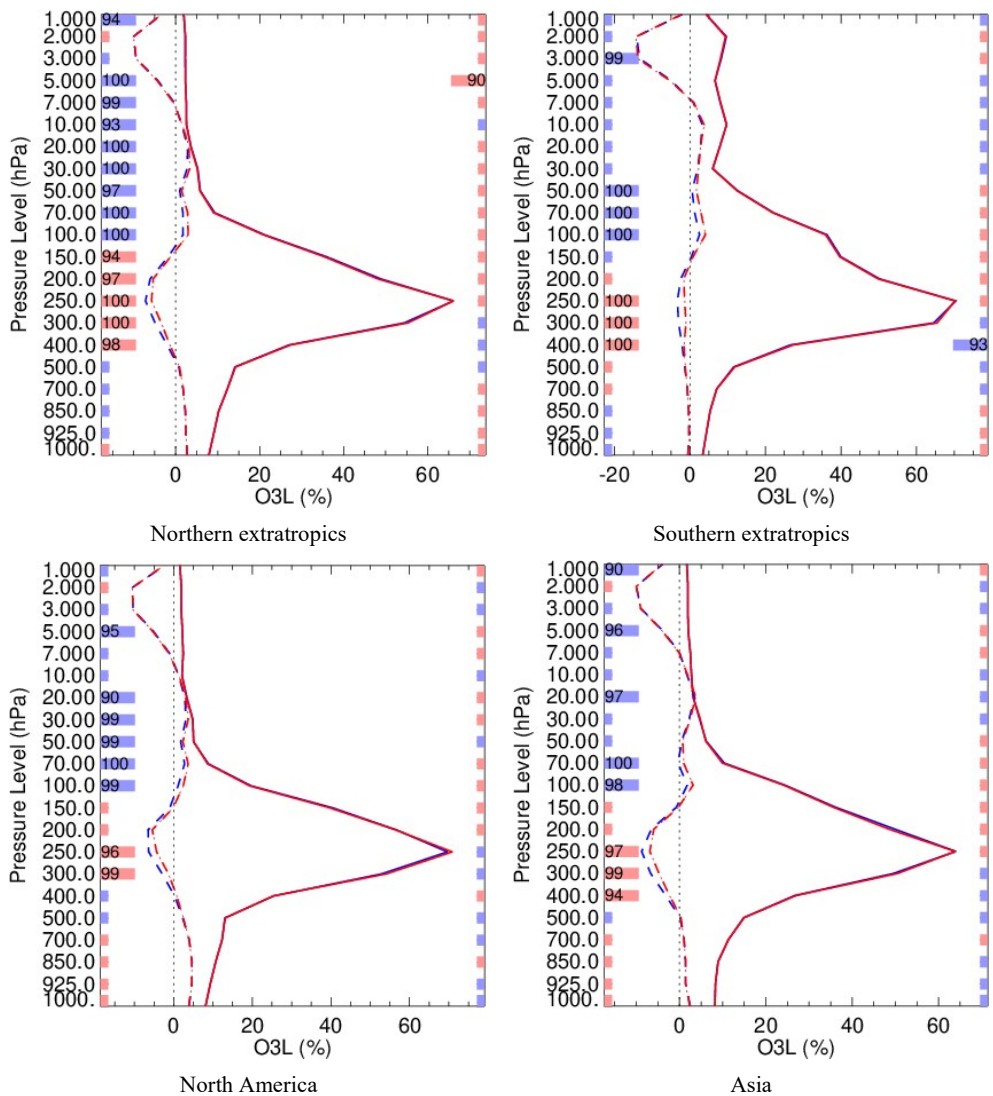



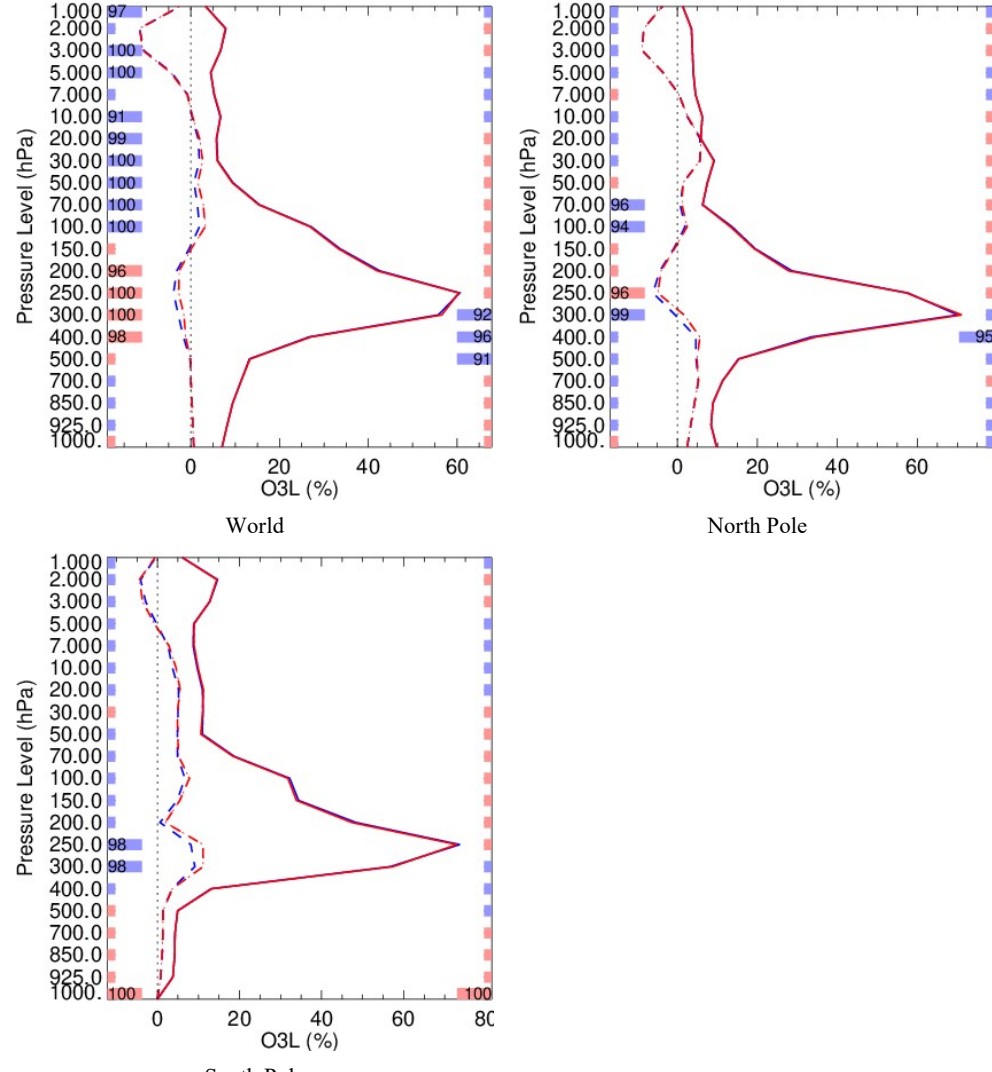

Figure 6. The bias (dashed line) and standard deviation (solid line) relative to GDPS analyses (%) for ozone forecasts at 240 hr lead-time from June 13th to August 31st 2019. Boxes on the left (right) indicate statistical significance levels for biases (standard deviation). Red (blue) boxes mean that the experiment which includes the sweep interpolation is better (worse). Small boxes mean that differences between both experiments are not statistically significant at the 90% confidence level.

210

Figure 7 shows the evolution of the bias and (Charron et al., 2012) standard deviation as a function of lead-time for the North America region at the 200 hPa level, which corresponds to region where both quantities vary the most abruptly along the vertical direction. The upper panel shows the standard deviation and bias errors over 10 days prediction. It demonstrates that sweep interpolation is characterized by the same error growth as the cubic interpolation. The figure also shows some

215    improvement in the bias error by using sweep interpolation. One possible explanation about this phenomenon is that lower order Lagrange interpolations tend to generate less oscillation and dispersion errors. To confirm this explanation, further study needs to be done which is out of the scope of this paper.

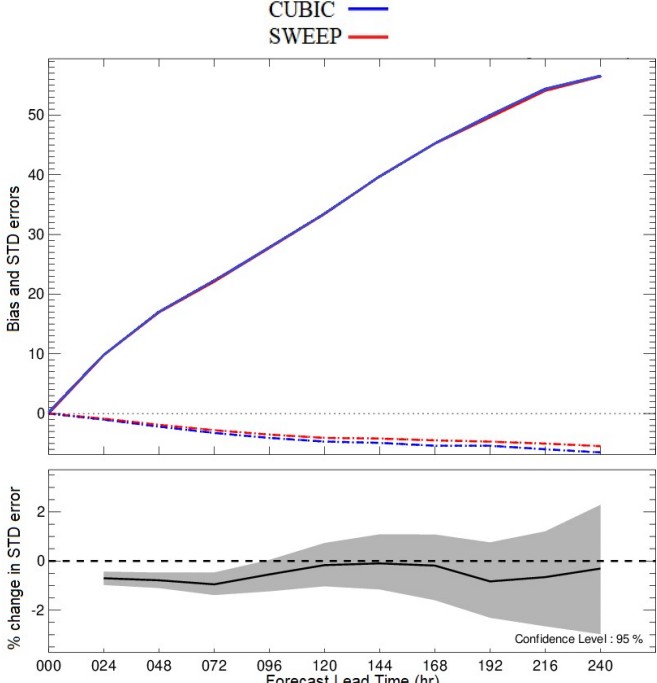

Figure 7. Standard deviation (solid line) and bias (dashed line) errors over North America for ozone at the level 200 hPa.

## 220    4 Impact of number of tracers on computational time

In the following, we investigate the impact of number of tracers on the timing of cubic interpolation process compared to sweep interpolation. We use the benchmark test case 2 (i.e., 24 hours forecasts of chemical $CH_4$ tracer) from the previous section and add additional tracers to the experiment. Note that in GEM we use cubic interpolation not only for advection solver, but also for exchanging the data between Yin and Yang grid (Qaddouri & Lee, 2011). In this study, we also



implemented sweep interpolation into Yin and Yang exchange of data. Figure 8 shows the impact of sweep interpolation on the timing of advection process, the exchange of data between the Yin and Yang sub-grids, and the total timing of the simulation. The numbers shown in the figure correspond to (time_cubic -time_sweep)/time_cubic x100. It is demonstrated that by increasing the number of tracers, percentage of computational time saving increases for the semi-Lagrangian operation, Yin and Yang exchange, and the total timing of the simulation. When using only 20 tracers, the performance of sweep interpolation is almost the same as cubic interpolation. However, when the number of tracers is increased to 230, sweep interpolation can reduce the computational cost of simulation in advection step by 25%. The sweep interpolation reduction in Yin and Yang exchange is about 18.5%, and its impact on the total timing of the simulation is about 16%.

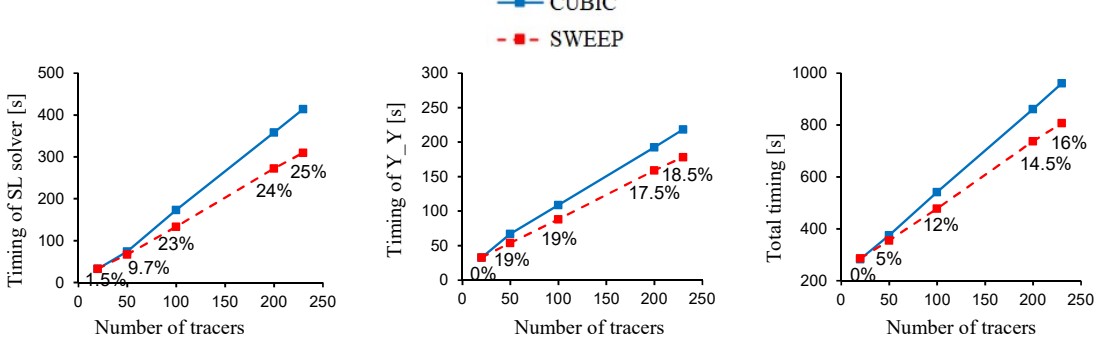

Figure 8. Computational time analysis based on the number of tracers.

## 5 Conclusion

In this study, we implemented the sweep interpolation scheme (an integration of two 3rd-order backward and forward interpolation steps) into ECCC's GEM model. Two theoretical benchmarks and a global ozone forecast experiment were used to investigate the performance of the model. Based on the results, sweep interpolation has a good agreement with cubic interpolation with lower computational cost.

The main benefit of using sweep interpolation in GEM is its lower computational cost compared to the cubic interpolation. By using this interpolation, we can reduce the timing of semi-Lagrangian interpolation, Yin-Yang exchange, and the total wall clock time of the simulation up to 25%, 19%, and 15%, respectively, for the experiments with around 250 tracers. Additionally, this replacement is very simple with minimum modifications required in the model. Sweep interpolation is suitable to be implemented in other NWP models, which also rely on a SL approach and polynomial interpolation schemes.

Although sweep interpolation is developed based on 3rd-order backward and forward Lagrange interpolation schemes, it can control the growth of error over the forecast lead-time comparable to the cubic interpolation. Ozone forecasting experiment using the global version of the GEM model shows that the sweep interpolation has a small impact on the ozone distribution along the vertical direction. The impact is particularly important in the tropopause region where transport processes play a

significant role. The overall impact of the new scheme on model biases and standard deviations was evaluated at different lead-times, which shows that the overall performance of the ozone forecasting system has not suffered from the use of a
faster interpolation approach. In some cases, we found that sweep interpolation can perform better and reduce the numerical errors. The impact on biases in ozone experiments generally increase with longer lead-time and is larger over some regions driven by transport processes. This improvement is likely related to the lower oscillations generated by lower order of Lagrange interpolation schemes used in sweep compared to the standard cubic interpolation, but this needs to be further investigated in a future study.

**Code availability.** GEM (Global Environmental Multi-scale), the Environment and Climate Change Canada's online weather prediction model, is a free software which can be redistributed and/or modified under the terms of the GNU Lesser General Public License as published by the Free Software Foundation. The GEM model is available to download from: https://github.com/ECCC-ASTD-MRD/gem/. The git repository contains the instructions that explain how to install the code, download the input data and run the model. The modified interpolation code can be downloaded from the Zenodo website:
https://doi.org/10.5281/zenodo.8246831. The model output requires a large amount of memory space in a binary format specific to Environment and Climate Change Canada's modelling systems. The conversion to other formats is possible by an email request to Ashu Dastoor (ashu.dastoor@canada.ca).

**Author contributions.** In this research work, MM developed the model, wrote the code, performed the simulation, and was responsible for analysis of the results, and writing the manuscript. JFC was responsible for writing the code, analysis the
results, and writing the manuscript. AD was responsible for the study concept, analysis of model simulations, and writing the manuscript. JDG and II were responsible for GEM-O3 set-up and error analysis of the model. AQ was responsible for writing the manuscript. All authors contributed to the editing of the manuscript.

**Competing interests.** The authors declare that they have no conflict of interest.

**Acknowledgements.** We thank our ECCC colleagues Rabah Aider, Andrei Ryjkov, and Kenjiro Toyota for their help with
running GEM and visualizing the results.

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
