# Peer review of "Sweep Interpolation: A Cost Effective Semi-Lagrangian Scheme in the Global Environmental Multiscale Model"

_EGUsphere, 2023_

## Author Comment (AC1)

Thank you for sharing your research. This paper presents a fourth-order accurate and cost-effective scheme called sweep interpolation, which uses fewer neighboring cells than the cubic interpolation. It significantly reduces computational time while maintaining very close accuracy to the typical fourth-order interpolation. However, there are still some issues that need to be addressed before it can be accepted for publication in GMD.

We greatly appreciate your interest in our work and your valuable and insightful comments. Your comments and suggestions have helped improve the manuscript.

(1) Different interpolation schemes should have different contents. Please compare the differences in contour maps between CUBIC and SWEEP interpolations in Figure 3, Figure 4, and Figure 5.

In response to your suggestion to include additional figures to compare the differences between CUBIC and SWEEP interpolations in Figure 3, Figure 4, and Figure 5, we have taken an alternative approach to address this issue. Instead of adding new figures, we have incorporated explanatory text within the paper to highlight the differences between the two interpolation schemes.

On page 7, we have included the following information: "For this case, the normalized infinity norm error ($E_\infty = \frac{\max |cubic-sweep|}{\max |cubic|}$)=0.001."

Page 8 now contains the explanation: "Here, the normalized infinity norm error is $E_\infty = 0.03$."

Page 9: "Here, the normalized infinity norm error is $E_\infty = 0.018$."

For your reference, we added the following figures to show the exact value of the tracers and the difference for the first case:

[Figure]

(2) In the atmospheric methane-like tracer test case, the differences between cubic and sweep interpolations are apparent (Figure 5b), and the reasons for these differences should be analyzed.

In response to your question about the differences between the cubic and sweep interpolations which are seen in Fig.5(b), we have added the following explanation of the sources of these differences:

"Although sweep interpolation was able to better control the mass error growth over the simulation time compared to the cubic interpolation for this case, it is not necessarily expected to perform better in all cases. Based on our discussion in the previous section, we expect sweep interpolation to provide almost the same accuracy as cubic interpolation. This is supported by Fig 5(b), which shows that sweep and cubic interpolations produce mass errors that are of the same order of magnitude. However, since both methods rely on different finite difference approximations, we expect to see differences in the evolutions of their respective error trends, which is confirmed by the results of Fig. 5(b)."

(3) Serial numbers are not marked in Figures 4 and 5.

Thanks for the comment. We have fixed the problem.

(4) Place all the pictures on one page in Figures 4 and 6.

In response to your request to place all the pictures on one page in Figures 4 and 6, we have removed the subfigure corresponding to the South Pole region from Fig6 and organized the remaining figures into two rows of 3 figures each for better consistency and presentation.

This modification aligns with your suggestion, and we believe it enhances the overall clarity and readability of our figures.

---

## Author Comment (AC2)

General comments:

This manuscript describes the application of the sweep interpolation with fourth-order accuracy in the GEM. As we all know, the interpolation algorithm for the velocity and tracer densities is vital for the semi-Lagrangian method. The authors proposed an elaborate idea of combining two interpolation stencils to cancel the leading errors. The sweep algorithm is efficient as the third-order one but with higher accuracy as the fourth-order one, and it is easy to implement. The numerical experiment results illustrate the efficacy of the sweep interpolation algorithm. I recommend the publication of this manuscript subject to a minor revision.

We would like to extend our sincere appreciation to the reviewer for the positive and constructive feedback on our manuscript. We are pleased to hear that our work on the sweep interpolation algorithm with fourth-order accuracy in the context of GEM has been well-received. We wholeheartedly agree with the reviewer's acknowledgment of the significance of the interpolation algorithm in semi-Lagrangian methods. To emphasize the importance of "sweep interpolation" in "semi-Lagrangian method", we have slightly revised the title of the paper to "Sweep Interpolation: A Cost Effective Semi-Lagrangian Scheme in the Global Environmental Multiscale Model". In the following, we address the reviewer's comments.

Specific comments:

It would be better that some details can be further explained:

- In 2D, there are four possible stencil combinations as shown in Fig. 1 of Mortezazadeh and Wang (2017). Is the selection of forward and backward interpolation stencils related to the parcel characteristic line? Or if the two stencils change according to the backward trajectory?

  The selection of the backward and forward only relates to the time step. As mentioned in the reference paper and the current manuscript, backward and forward interpolation are used in successive time steps, otherwise the truncation error won't be cancelled every two time steps. For 2D cases, there isonly onepossible stencil combination, backward in x and y directions, and then forward in x and y directions, and this combination is not related to the parcel characteristic line or position.

- The description of the tests is too brief, such as sec. 3.1. Please add more information, such as what the spatial resolution is?

In response to the request for a more detailed description of the tests, particularly in Section 3.1, we have made the necessary additions to enhance the clarity of the paper.
On page 6, we have included additional information about the flow field and grid resolution: "The flow field utilized in this benchmark is positive definite. The spatial resolution used in both horizontal directions is approximately 105 [km] and a time step of 7200 [s] is employed, which yields a maximum value of Courant number of 0.85426."

Furthermore, on page 8, we have added the following: "The horizontal spatial resolution and time step used in this example are, respectively, 205 km and 3600s, which yields a horizontal Courant number (CFL) of 5.0."

- Why the total mass of sweep scheme is decreasing, while cubic scheme is increasing in Fig. 5.

In response to the above question regarding the differences between cubic and sweep interpolations in the atmospheric methane-like tracer test case (Figure 5b) and the reasons behind these differences, we have added the following explanation:

"Although sweep interpolation was able to better control the mass error growth over the simulation time compared to the cubic interpolation for this case, it is not necessarily expected to perform better in all cases. Based on our discussion in the previous section, we expect sweep interpolation to provide almost the same accuracy as cubic interpolation. This is supported by Fig 5(b), which shows that sweep and cubic interpolations produce mass errors that are of the same order of magnitude. However, since both methods rely on different finite difference approximations, we expect to see differences in the evolutions of their respective error trends, which is confirmed by the results of Fig. 5(b)."

Technical corrections:

L77: The variable staggering in the vertical direction is the Charney-Phillips grid, so it should not be the Arackawa-C grid in the vertical direction.

To answer this comment and to provide further clarification, we modified the paper accordingly:

"The governing equations are formulated using spherical coordinates together with a log-hydrostatic pressure type terrain following vertical coordinate (Husain et al., 2020). They are discretized on an Arakawa C grid (Arakawa, 1988) in the

horizontal, whereas in the vertical direction, they are discretized using a Charney–Phillips grid. Tracer transport is accomplished by first solving the advection equation for a passive tracer and then by adding contributions from physics forcings in split mode. The current interpolation scheme in GEM is fourth-order-accurate cubic Lagrange interpolation. It is used to calculate the variables at the departure point, as well as to perform the exchange of data on the boundaries of the two sub-grids of the global Yin-Yang grid. In this study, we document the impact of using sweep interpolation for the advection of tracers as well as for the exchange of data between Yin and Yang subgrids in GEM."

L83: "sub grids" to "subgrids"

Thank you for the comment. We have modified the word accordingly.

L116: " ,on 1D" to ", along the 1D direction"?

Thanks, the modification has been applied.

---

## Author Comment (AC3)

This work extends the research conducted by Mortezazadeh and Wang in 2017. In this study, the sweep method is further validated through a series of idealized tests, including 2D vortex simulations and Hadley-like meridional circulation, as well as an Atmospheric methane-like tracer test and global model forecasts. This manuscript demonstrates that the sweep method can significantly reduce computational costs by approximately 15% without compromising accuracy. This improvement is achieved by implementing two 3rd-order backward and forward polynomial interpolation schemes over two consecutive time steps, as opposed to using a 4th-order interpolation method. The results presented in this manuscript are intriguing and robust, supporting its acceptance for publication with only minor revisions.

We would like to extend our sincere thanks to the reviewer for the positive and insightful comments regarding our manuscript. We are pleased that the reviewer acknowledges the extension and validation of our work.

1. In Figure 3(b) and 4(b), the black dotted line represents the mass error after every two time steps. To enhance clarity, consider changing the label from "Error after two time steps" to "Error every two time steps."

   Thanks for the comment. We have replaced "Error after two time steps" with "Error every two time steps" in the revised manuscript.

2. It is essential to delve into the motivation behind and the conclusions drawn from the 2D vortex tests. Further discussions are warranted to provide a comprehensive understanding of their significance.

   Here to clarify the main motivation behind using the 2D case has been explained and added to the paper (page 7):

   "The main reason of choosing the 2D case was showing the oscillation in the mass error for sweep interpolation. In this case, the oscillation is obvious and helps explain the behavior of sweep interpolation. The same behavior has been seen in the other test cases (see next sections). For this case, the normalized infinity norm error ($E_\infty = \frac{\max |cubic-sweep|}{\max |cubic|}$)=0.001."

   Further explanation about this case has been provided into the next comment.

3. Figure 3a illustrates that the error distribution of the 2D vortex simulations is less noisy outside the vortex region when using the sweep method. Is this observed reduction in noise attributed to the method's capability to minimize

dispersion and dissipation errors? Additional clarification on this matter would be beneficial.

As we discussed in Section 3.4, we did observe an improvement in bias error when utilizing the sweep interpolation method. One plausible explanation for the observed reduction in noise, particularly outside the vortex region in Figure 3a, is that lower order Lagrange interpolations, as employed in the sweep method, tend to generate fewer oscillation and dispersion errors. While this is a plausible explanation, we must acknowledge that confirming this hypothesis would require further in-depth investigation, which falls beyond the scope of the current paper. We intend to explore this in our future research endeavors.

To provide greater clarity, we have incorporated the following explanation into the manuscript:
"Figure 3(a) shows that the error distribution associated with the sweep interpolation is less noisy compared with the cubic interpolation error, especially outside the vortex region, which could be explained by the fact that the lower order Lagrange interpolation used in the sweep algorithm generates less spurious oscillations compared to the standard cubic interpolation."

4. Section 3.3, pertaining to the Atmospheric methane-like tracer experiment, lacks a clear description of the experiment's design. While information on resolution and time steps is provided, it would be beneficial to include more detailed descriptions.

We appreciate reviewer's feedback. In this specific experiment, there are no physics or chemical production and sinks, which is why there are limited additional details to provide. For improved clarity, we have updated the section to include the following description:

"In this test case, we compare 48-hour forecasts of atmospheric methane (CH4) like passive tracer (without chemical productions and sinks) using sweep interpolation and cubic interpolation. These experiments were performed with the global version of GEM NWP model using a 30-minute time step and 105 [km] horizontal resolution resulting in a maximal courant number of 4.7. The height of the model top was chosen to be at 0.1 hPa and 84 vertical levels were used. The vertical grid resolution is non-uniform as a result of the choice of vertical coordinate which is based on the logarithm of the hydrostatic pressure (Husain et al., 2020). The methane-like experiment was initialized from a climatology based on a multi-year simulation performed with the GEM model. The model employs a simplified approach, in which methane production and loss are

predetermined based on present-day conditions (Prather et al., 2012). Figure 5(a) presents meridional cross sections of CH4 at the end of day 1. Solutions from both interpolators look qualitatively the same, and the sweep interpolation provides acceptable results in comparison with cubic interpolation. Figure 5(b) shows the mass error over 24 hours. It shows that both cubic and sweep interpolations could control the error and keep its range below 0.005% after 48 time steps of simulation. Here, the normalized infinity norm error is $E_\infty = 0.018$."